# ICMR task force project- survey of the incidence, mortality, morbidity and socio-economic burden of snakebite in India: A study protocol

Jaideep C. Menon 1☯*, Omesh K. Bharti 2☯, Rupinder S. Dhaliwal3‡, Denny John 4☯, Geetha R. Menon5☯, Ashoo Grover 6‡, Joy K. Chakma7☯

1 Preventive Cardiology & Population Health Sciences, Amrita Institute of Medical Sciences & Research Centre, Amrita Vishwa Vidyapeetham, Kochi, Kerala, India, 2 Department of Health & Family Welfare, Shimla, Himachal Pradesh, India, 3 Non-communicable Diseases Division, Indian Council for Medical Research (ICMR), New Delhi, India, 4 Department of Public Health, Amrita Institute of Medical Sciences, Amrita Vishwa Vidyapeetham, Kochi, Kerala, India, 5 National Institute of Medical Statistics- ICMR, Indian Council of Medical Research, New Delhi, India, 6 Research Methodology Cell, Non Communicable diseases, Division, Indian Council of Medical Research, New Delhi, India, 7 Non-Communicable Diseases Division, Indian Council of Medical Research, New Delhi, India

☯ These authors contributed equally to this work.
‡ RSD and AG also contributed equally to this work.
* menon7jc@gmail.com

**Data Availability Statement:** Annexures used in this protocol are available in public repository @John, Denny; Menon, Jaideep (2021): ICMR –

# Abstract

## Background

Snakebite is possibly the most neglected of the NTDs (Neglected Tropical Diseases). Half of the global deaths due to venomous snakebites, estimated at 100,000 per year, occur in India. The only representative data on snakebite available from India is the mortality data from the RGI-MDS study (Registrar General of India- 1 Million Death Study) and another study on mortality from the state of Bihar. Incidence data on snakebite is available for 2 districts of the state of West Bengal only. Hospital-based data on snakebite admissions and use of ASV are gross underestimates as most snakebite victims in rural India depend more on alternate treatment methods which do not get represented in National registries. The proposed study is a multi-centric study to determine the incidence, morbidity, mortality and economic burden of snakebites in India covering all 5 geographical zones of the country.

## Protocol

A community level surveillance for snakebite covering 31 districts in 13 states of India in order to obtain annual incidence of snakebites from the community. Frontline health workers will be trained to gather information on new cases of snakebite over the study period of 1-year, from "wards" (smallest administrative subunit of a village or town) that they represent in the study districts. Dedicated field officers would collect data on snakebites, victim characteristics, outcomes, utilization of health facilities on a questionnaire sheet designed for this purpose. The study duration is for 18 months from April 2022 to October 2023.

Task Force Study on the incidence, mortality, morbidity and socio-economic burden of Snakebite in India: A study protocol. figshare. Dataset. https://doi.org/10.6084/m9.figshare.14072108.v1.

**Funding:** The study is funded as a National Task Force project of the Indian Council of Medical Research (ICMR), Government of India. The PI for the study is the corresponding author. The Grant Number is RFC No. NCD/NTF/16/2019-29 dated 20.02.2020. The ICMR has no role other than for funding the project.

**Competing interests:** The authors have no competing interest to declare with regards to the present project/manuscript

## Discussion

The study would be the first of its kind in India looking prospectively at the incidence of snakebite covering 13 states in 5 zones of India and a population of 84 million. Our study covers 6.12% of the total population of the country as compared to the incidence study conducted in Sri Lanka which covered 1% of the total population.

## Introduction

Snakebite is the most neglected of all tropical diseases. In fact, it is only in 2017 that snakebite was added back onto the WHO (World Health Organization) list of neglected tropical diseases (NTDs), after being struck off the list in 2013 [1]. Geographically, the greatest impact of snakebite is in the tropical and subtropical regions, with the highest occurrence in India. Global estimates of snakebite range from 4.5 million to 5.4 million bites annually with an estimated 2 million of them in India with significant physical, mental and socioeconomic consequences [2]. As per the Registrar General of India- Million Death Study (RGI-MDS) the number of deaths due to venomous snakebite in India is 46,900 per year [3]. This is considerably high, compared to only 10–12 deaths per year, due to venomous snakebite in the US and Australia, this despite the fact that less populous Australia has probably more venomous species [4]. Reports suggest that only 20–30% of victims of snakebite in rural India seek treatment in hospitals [5–7].

Under reporting and lack of data on incidence, mortality and socio-economic burden make it difficult to understand the true impact of a condition on the health of a population. The largest study that provides mortality data for snakebites is the Cause of Death survey by the RGI [3]. There is paucity of data on the incidence of venomous snakebite in India, barring two surveys in the South 24 Parganas and Burdwan districts of West Bengal where 4871 cases were reported in 2 years from a population of 1.9 million [5]. More recently a study on mortality related to snakebite was done in the state of Bihar [8]. A study using information on compensation paid to victims of snake bite mortality or morbidity from 2012–2015 by the Forest Divisional Office in Kannur concluded that only 58% of the identified cases appeared to have received compensation [7]. Most of the other available data is from hospital based case series from Maharashtra, West Bengal, Kerala and Andhra Pradesh [9, 10]. Vaiyapuri et. al. (2013) using a household survey in rural Tamil Nadu demonstrated significant reduction in medium and long-term family income due to snakebite additional to the immediate costs of a bite [11]. Studies conducted in low- and middle-income countries, such as Nigeria, Sri Lanka and Sub-Saharan Africa have reported economic burden of snakebites using Disability Adjusted Life Years (DALYs) [12]. However, no studies reporting DALYs on snakebite have been reported in India [12].

The proposed study aims to address some of these issues by conducting an annual incidence study to provide the epidemiological and socio-economic burden due to snakebite. It will be conducted in 13 states located in 5 different geographical zones of the country and will provide state specific estimates on incidence, mortality, pattern of injuries, treatment seeking behaviour and cost of illness among snakebite victims.

### Rationale for the study

Reliable epidemiological and socio-economic data on snakebites is essential to regulate control of anti-venoms and their distribution policies in a country. The epidemiology and economic

data on snakebite is also essential for advocacy, recognition, and fund allocation by the Government for the mitigation of snakebite in India.

## Methods

### Study aims

Our study aims to estimate the epidemiological and socio-economic due to snakebite in India The objectives of the study are to:

*Objective 1*- Determine the incidence and mortality due to snakebite in the study districts

*Objective 2*- Document the clinical course- predominant symptoms, duration of hospital stay, ASV vials used, complications etc in victims admitted and treated for snakebite

*Objective 3*- Understand the treatment seeking behaviour of the snakebite victims

*Objective 4*- Determine the cost of illness due to snakebite in the community

*Study design* -Cross-sectional

The cross-sectional population in this case is the study population comprising all resident individuals in the study district.

Outcome measured is the event of snakebite in the population. The event of a snakebite is recorded within two weeks of the bite, death due to snakebite or discharge from hospital, irrespective of whether the victim was admitted in hospital, treated by alternate healers or even in case the victim did not seek treatment. Frontline health workers (ASHAs), who serve a population of 1000 individuals (ward level), and are generally aware of health events in the communities they serve.

The study has been modelled along the lines of a previous Sri Lankan study which covered the whole of the island surveying <1% of the total population [13]. The study districts have been identified to be truly representative of the state with regards to rainfall, altitude, and foliage, which in turn determines the native species of snake.

Based on the topographical location of the sites, 1–4 districts from 13 states representing all the geographical zones of the country have been selected for the study. All cases of snakebite occurring in the study district during the study period of 1 year would be recorded. A total of 336 blocks from 31 districts covering a total population of approximately 83.9 million will be included in the study (Table 1). All episodes of snakebites in this population will be prospectively recorded for a period of one year.

### Participants

**Inclusion criteria.**

i. Victims and family members referred by Accredited Social Health Activist (ASHAs) to the Field Officer

ii. Those providing consent to participate

### Exclusion criteria

i. Symptoms related to poisoning or non-ophid bites

ii. Refusal to consent

**Table 1. Included districts, population and topography.**

| Zone | State | Disricts | Taluks | Population | Topography |
|---|---|---|---|---|---|
| West | Maharashtra | Raigad | 15 | 2,966,109 | Coastal |
| | | Pune | 13 | 10,617,513 | Highland |
| | | Nanded | 16 | 3,784,815 | Semi-arid Deccan plateau |
| Central | Rajasthan | Jaisalmer | 3 | 756,681 | Dry arid desert |
| | | Udaipur | 11 | 34,55,041 | Highland |
| | | Bikaner | 8 | 2,363,937 | Predominantly desert |
| South | Kerala | Ernakulam | 15 | 3,695,969 | Western Ghat section |
| | | Kannur | 7 | 2,840,901 | Western Ghat section |
| | Tamil Nadu | Tiruchirapalli | 14 | 3,065,298 | Predominantly Highland |
| | | Tiravanur | 14 | 4,197,845 | Coastal |
| | Andhra Pradesh | Nellore | 5 | 3,336,965 | Coastal |
| | | Chitoor | 3 | 4,699,996 | Hilly Highland |
| East | Odisha | Cuttack | 25 | 2,955,153 | Coastal |
| | | Sambalpur | 20 | 1,172,277 | Highland |
| | West Bengal | Bankura | 22 | 4,049,855 | Forest and Highland |
| | | East Burdwan | 31 | 8,696,844 | Riverine plain |
| | | Jalpaiguri | 7 | 4,360,824 | Sub-himalayan Terai |
| | | South 24 Paraganas | 5 | 9,190,368 | Mangrove forest |
| North | Himachal Pradesh | Kangra | 15 | 1,700,344 | Sub-Himalayan plains, lowlands, and highlands |
| | | Una | 5 | 586,841 | Sub-Himalayan plains, lowlands, and highlands |
| | | Chhamba | 7 | 584,484 | Sub-Himalayan plains, lowlands, and highlands |
| | Uttarakhand | Nainital | 8 | 1,074,885 | Hilly and forest |
| North-East | Arunachal Pradesh | Papum Pare | 15 | 1,98,821 | Hilly and forest |
| | | Pakke Kessang | 5 | 88,605 | Hilly and forest |
| | Meghalaya | East Khasi Hills | 11 | 929,988 | Hilly and forest |
| | | West Garo Hills | 7 | 7,24,346 | Hilly and forest |
| | Mizoram | Aizawl | 5 | 400,309 | Hilly and forest |
| | | Lunglei | 4 | 161,428 | Hilly and forest |
| | | Champai | 4 | 125,745 | Hilly and forest |
| | Tripura | Dhalai | 8 | 425,887 | Hilly and forest |
| | | South Tripura | 8 | 986,377 | Hilly and forest |
| Total | 13 | 31 | 336 | 83,977,167 | |

## Ethical approval

Participants will be informed about the nature of the study and will be assured that privacy will be maintained, and information provided by the respondent will be held confidential and only be used for research purposes. Their willingness to participate will be sought and informed written consent will be taken before including them in the study. Social and cultural values of the participants will be respected and considered as needed. Information obtained during research will not be used for any other purpose except research and research findings will be disseminated honestly. The protocol has been developed using the STROBE guidelines for cross-sectional studies.

Ethics approval has been obtained from the Institutional Ethics Committee of Amrita Institute of Medical Sciences (NTCC), Kochi [IEC-AIMS-2019-CARD-169, dated 3rd October 2019] and leading implementing institutes/centres and or endorsement/administrative concurrence is obtained from the participating State government health department. The study has been registered with the CTRI- Clinical Trial Registry of India.

**Table 2. Estimated snakebite cases and deaths for a period of 1 year.**

| Event | Expected frequency | Total population | Expected numbers |
|---|---|---|---|
| Bites | 0.154% | 83,977,167 | 129,325 |
| Deaths | 0.006% | 83,977,167 | 5039 |

## Sample size

Based on the topographical location of the sites, 1–4 districts from 14 states representing all the geographical zones of the country have been selected for the study. All cases of snakebites occurring in the study district during the study period of 1 year would be recorded. A total of 336 blocks from 31 districts covering a total population of approximately 83.9 million will be included in the study (Table 1).

All episodes of snakebites in this population will be prospectively recorded for a period of one year. Based on the estimated population of 83.9 million and 16.78 million households in Census of India 2011, we aim to identify snakebite cases and snakebite deaths for a period of 1 year.

An estimate of the percentage of bites and deaths has been obtained from the District 24 Parganas of West Bengal [3] assuming these estimates as prior estimates the expected number of bites and deaths in the target population is as below in Table 2.

## Outcomes

The study will measure annual incidence, mortality, and treatment costs of snakebites across 13 states and 31 districts across India. The specific primary and secondary outcomes are listed below.

### Primary outcomes

- Incidence of snakebites in respective topographical areas

- Incidence of deaths due to snakebites in respective topographical areas

- Socio-economic burden of snakebite (using cost of treatment and DALYs) and its correlates

### Secondary outcomes

- Estimation of DALYs lost due to snakebite in India

## Statistical analysis

The study is proposed to estimate the incidence rates and socio-economic burden for the target districts. The population in the selected districts will be covered for a period of one year to obtain the new snakebites cases and deaths.

**Epidemiological components.** Population based incidence rates of snakebite with 95% confidence intervals will be presented at national and state levels. Population based incidence rates will be calculated using the "Survey" package in R programming language. Individual level variables (e.g. age, sex) will be considered only for descriptive analysis. The explanatory variables for snakebite incidence will include population density, gender, occupation, education, income, and climatic zones. The categorical variables will be presented in the form of

frequencies and percentages and the continuous variables will be presented as means and standard deviations.

Population based incidence rates of snakebite with 95% confidence intervals will be presented at national and state levels. The following measures will be computed:

$$\text{Incidence risk} = \frac{\text{Number of incident cases of snakebites in the time period X } 100,000}{\text{Population at risk (population of the district)}}$$

$$\text{Mortality risk} = \frac{\text{Number of deaths in the time period X } 100,000}{\text{Population at risk (population of the district)}}$$

$$\text{Case}-\text{fatality rate} = \frac{\text{Number of deaths from snakebites in the time period X } 100000}{\text{Number of new cases of snakebites in the time period}}$$

Above mentioned epidemiological parameters will be computed at district level along with age and gender distributions for each state.

**Economic burden.**   Cost of treatment: The median out-of-pocket cost of different cost elements (direct medical and non-medical and indirect) will be estimated based on the data reported by the victims or a household member. We will use cost of treatment episode to calculate direct medical and non-medical costs. The total sum spent by patients for a particular cost item and the proportion of the patients that incurred that cost will be applied to the national/state incidence of snakebite to estimate the total annual out-of-pocket cost of snakebite at the national/state level.

DALYs will be calculated using the following formula: DALY = YLD + YLL. The template developed by the World Health Organisation will be used for estimation of DALYs. Population data from the 2011 national census will be obtained from the Registrar General of India. For envenoming we will use disability weights for poisoning from GBD study including that of National Burden Estimates. DALYs will be computed separately for males and females at the national level.

All data collected in the questionnaire will be analyzed using appropriate statistical techniques. Data analysis will be performed in Stata version 15 and in R programming language version. The categorical variables will be presented in the form of frequencies and percentages and the continuous variables will be presented as means and standard deviations.

**Data collection, quality checks and monitoring.**   Each state will be supervised by a Principal Investigator. Each selected district will have a District Coordinator for supervision and monitoring of data collection. The district coordinator will have field workers for every 10–15 blocks who will closely work with ASHAs- (Accredited Social Health Activist) for getting information on a new case of snakebite. ASHAs are frontline health workers at the community level working as the eyes and ears of the health system with an ASHA to a ward (smallest administrative subunit of the village or town). An ASHA typically represents a population of 1000 and is generally aware of all the health events occurring in their ward. The duties of ASHAs include and not limited to immunisation and family planning, facilitating directly observed therapy short-course (DOTS) therapy for tuberculosis at ward level, surveillance, and monitoring for communicable diseases- polio, vector borne febrile illnesses etc, surveillance for NCDs at the ward level, sanitation related activity, participation and facilitation of state or district specific activities at the community level.

We are aware that there would be instances where ASHAs would might miss some of the snakebites that occur. These will the minimized to a large extent through the following; training sessions for ASHAs for identification of any warning symptoms in case of a bite, do's and

dont's following a bite, and identification of the common venomous snakes of the region. These training sessions will also include use of Google Forms using tablets for ASHAs to enter the data as mentioned in S2 File. Additionally, through the grant, we have also kept a provision for incentivising the ASHAs for reporting each snakebite case to the District Coordinator.

As soon as a new case is reported by the ASHAs in their ward the field worker will visit the household and fill the questionnaire. Written consent will be taken from the subject (S1 File). ASHAs will be given incentive for reporting any information related to incidence or death due to snakebites in that area. The field worker will collect the data regarding the bite, treatment and outcomes from the victim within 2 weeks of bite, or discharge from hospital, or from family members in case of death. Information about the victim, profile of envenomation and complications thereof, other related characteristics, treatment outcome and any other related details, and medical attention seeking behaviour pattern among the bitten (preference for modern vis-à-vis alternate systems of medicine) would be gathered from the questionnaire sheet. Details regarding treatment, complications and outcomes would be as gathered from the discharge / death summary of hospitalised patients. In cases where the discharge/death summaries are incomplete, the District Coordinator will be informed who will coordinate with the Principal Investigator of the state to reach out to the hospital/doctor to receive all relevant details. For the cases where the victim has sought native or local treatment, this will be documented by the ASHAs based on discussions with the victim, and will be supplemented by any additional information by the District Coordinator wherever needed.

Data from the questionnaire will be checked for completeness and consistency. It will be entered by the district coordinator in real time data entry software on a tablet and will be synchronised to the Central server located at technical coordinating centre at AIMS Kochi. (Fig 1). Random verification of data entered by the ASHA and Field officer would be done by the district coordinator and Principal Investigator of the district by telephoning the victim/ family. At the end of the interview with the victim/victim's kin, the data collector shall conduct a brief awareness/sensitization session for the local villagers using pre-designed learning tools like pamphlets and other educational materials that are available with the investigators. An easy, comprehensible first-hand material on "Do's and Don'ts for Snakebite" (translated in local vernacular) along with photographs of the common venomous snakes/nonvenomous snakes of the area shall be displayed and disseminated among the mass at all community levels.

Each investigator will get access to only their site's data with the help of independent user login details. The overall access to the data from all centres will be at ICMR-NIMS and AIMS Kochi. The data quality in terms of data completeness, accuracy and consistency will be checked by ICMR-NIMS team in close cooperation with the technical coordinating unit at AIMS Kochi. Real time data from each site will be displayed in the dashboard that will be designed for this purpose.

The study would be facilitated through the offices of the DPM-NHM of the district and the district medical administration (CMO/DMO) of the district.

However, for information related to any family member whose age is below 18 years, the mother or the father shall be interviewed. Information about the victim, profile of envenomation and complications thereof, other related characteristics, treatment outcome and any other related details will be noted but kept coded and confidential. Fig 1 provides the flowchart of study management process.

Informed consent from all participants will be taken prior to administration of any questionnaire (S1 File). A pre-specified questionnaire developed for the research study will be used to collect data regarding socio-demographic, hospitalisation, and economic details (S2 File). The questionnaire has been developed after consultation with experts working in the field of snakebites in the country and discussion among authors.

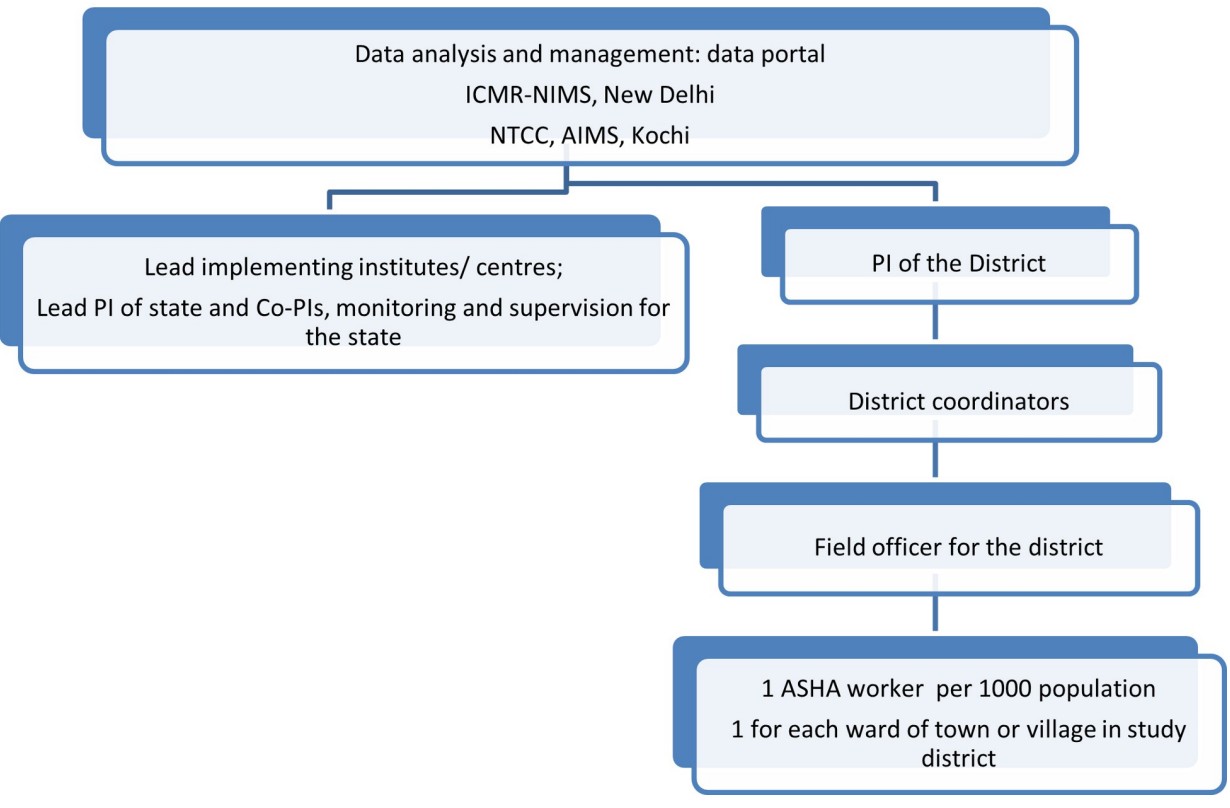

**Fig 1. Flowchart of study management process.**

All documents administered to the study participant (S1 and S2 Files) will be translated to the local language of the state. We will use a single translator who has knowledge of English and Local Language. Once the English version is translated to the target language, the local State Principal Investigator will validate the translation (State Principal Investigator for our study are familiar with the local language of the state). The translated version will be then translated back to English language. The state Principal Investigator will then compare the original and back-translated version and along with the translator will prepare the final draft of the questionnaire which will be used to be administered in the field.

## Data entry and storage

One of the co-authors will review the data entry to check for any discrepancies including any data entry errors from the data entry form. The data will be stored in a desktop computer with access to the Overall Study Coordinator and Principal Investigator (JCM). Once the data entry is completed and cleaned, the data sheet will be transferred to the laptops of the co-authors (JCM, GRM & DJ) for further analysis. After analysis these data sheets will be destroyed in these laptops and the data sheet would be available only with the desktop present at the research unit of AIMS/ICMR-NIMS.

Data gathered at the Panchayat (village) level would be collated on the Questionnaire sheet in the regional language by the field officers, gathered data is then entered on the Tab PC by the District coordinator in English using the software provided, from where it would be synced on to the server at AIMS/ICMR-NIMS, accessible only to JCM, and GRM. De-identified

details would be used for statistical analysis and reporting. Details gathered would not be shared on any public domain and would be kept confidential.

## Study challenges

Snakebite is generally a disease of the working community, being most common among farmers, rubber tappers, tea/coffee estate pickers, brick kiln workers, and plywood industry workers. Most bites are accidents, which occur at the workplace or at home with the lower socioeconomic groups being disproportionately affected. There could be a recall bias in victims especially in the case of mild degree of envenomation, non-venomous or dry bites.

## Study limitations

The present study extrapolates data from 13 major states throughout the country. However, in each state, 1–4 districts are selected taking into consideration the topography, trying to compare bites and species between a dry vs wet district,coastal vs hilly district, this because biting species tend to differ with topography, aridity, foliage and may not be totally representative of the whole country. Likelihood of other bites like rat and other insect bites being mistaken for snakebite especially when there has not been any envenomation.We also envisage recall bias on expenditure for management of snakebite reported by study participants.

## Distribution of study results

The study results will be submitted to a suitable peer-review publication within 6 months of study completion. Additionally, the results will also be presented in suitable national/international conferences based on resources available for participation. The study results will be presented using STROBE guidelines for cross-sectional studies.

## Study status

The study was approved and funding provided by the Indian Council of Medical Research. Software for the project has been written and is being field tested through a small pilot. The Tab PCs have been procured and been loaded with the software and are in the process of being shipped to the different study districts. Training sessions for data entry have been completed for the states of Kerala and Rajasthan.

## Conclusion

Our study would be the first of its kind in India where we would look prospectively at the incidence of snakebite across 13 states and 5 zones of India covering 84 million populations. Our covers 6.12% of the total population of the country compared to the incidence study conducted in Sri Lanka which covered 1% of the total population.

This study looking at the incidence of snakebite in selected districts of 13 states in India (in Phase 1), would help fill-in a notable lacuna of paucity in data on venomous snakebite in India. Snakebite is a problem neglected by the governmental agencies, foreign funding bodies, health care professionals and the media alike. There is one snakebite death for each 2 deaths from HIV, in sharp contrast there are hardly any funds available globally for research in snakebite while HIV is a much-researched subject. Snakebite is an occupational hazard in most situations where if correctly diagnosed and treated the victim would return to as productive a life as before, which cannot be said of most other NCDs. Despite this snakebite remains neglected because it is a disease of the socioeconomically marginalised groups that tend not to be in the spotlight.

This study is expected to give the much-needed incidence of snakebite in the country that would help the government in planning for this disease in the future by way of allocation of medicines and health resources.

## Supporting information

**S1 File. Participant information sheet and consent form.**
(DOCX)

**S2 File. Study questionnaire.**
(DOCX)

## Author Contributions

**Conceptualization:** Jaideep C. Menon, Omesh K. Bharti, Rupinder S. Dhaliwal, Denny John, Geetha R. Menon, Ashoo Grover.

**Data curation:** Jaideep C. Menon, Denny John, Geetha R. Menon.

**Formal analysis:** Denny John, Geetha R. Menon.

**Funding acquisition:** Jaideep C. Menon, Omesh K. Bharti.

**Investigation:** Jaideep C. Menon.

**Methodology:** Jaideep C. Menon, Omesh K. Bharti, Rupinder S. Dhaliwal, Denny John, Joy K. Chakma.

**Project administration:** Jaideep C. Menon, Rupinder S. Dhaliwal, Joy K. Chakma.

**Resources:** Rupinder S. Dhaliwal, Joy K. Chakma.

**Supervision:** Jaideep C. Menon, Omesh K. Bharti, Joy K. Chakma.

**Validation:** Jaideep C. Menon, Denny John, Geetha R. Menon.

**Visualization:** Jaideep C. Menon.

**Writing – original draft:** Jaideep C. Menon, Omesh K. Bharti, Denny John.

**Writing – review & editing:** Jaideep C. Menon.

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
