## [Decision Letter · Decision Letter 0]

15 Jul 2021

PONE-D-21-07087

ICMR Task Force Project- Survey of the incidence, mortality, morbidity and socio-economic burden of snakebite in India: A study protocol

PLOS ONE

Dear Dr. MENON,

Thank you for submitting your manuscript to PLOS ONE. After careful consideration, we feel that it has merit but does not fully meet PLOS ONE’s publication criteria as it currently stands. Therefore, we invite you to submit a revised version of the manuscript that addresses the points raised during the review process.

We look forward to receiving your revised manuscript.

Kind regards,

Rakhi Dandona

Academic Editor

PLOS ONE

Additional Editor Comments:

1. The manuscript is not written as a study protocol but reads like an internal/study document. Other reviewers have also referred to this issue with the writing style. Please update.

2. Introduction, second last paragraph - The premise of undertaking the study is mentioned as the need to “understand the socioeconomic impact and to regulate control of anti-venoms and their distribution policies.” The evidence so far from India suggests that anti-venom is not used in many cases, its availability is a concern in rural areas, and not all victims are taken to a medical doctor. More in-depth exploration is needed for understanding the issues with anti-venom use, but that is not covered or addressed in this protocol.

It appears that the investigators have assumed that most victims are taken to hospital, as reflected in Annexure 2 (proforma) – question 17. If the victim was not taken to a hospital, then it appears no further information on injury/treatment is collected.

3. Introduction, last paragraph – There are reasonable data available to suggest that snakebite mortality is a public health concern, and undertaking a study for incidence and mortality seems not fully justified. Much less data are available on the pattern of injuries and cost of illness among snakebite victims, however, the study proforma does not show that it will be captured well for meaningful interpretation. Annexure 3 does not capture much on injury pattern and assumes that injury will be seen in all victims, and that the respondents will be able to inform injury details. Furthermore, the economic detail section also assumes that all victims will be income-earners and does not differentiate between a fatal and non-fatal case. The investigators could refer to injury-related economic loss/details guidelines to capture relevant information.

4. Secondary outcomes (page 12) - Annexure 3 does not capture much information on management pattern for venomous snakebites that can help strengthen the national protocol. It also does not capture information on medical attention seeking behaviour pattern among the snakebite victims beyond hospitals.

5. Methodology, data collection (pages 14-16)

a. How will snakebite cases be identified by the field workers? Will this be based on hear-say from the local people or victim’s family? There is enough evidence to suggest that a snake is not seen in all cases of snakebites. How will then confirmation be made of a case being that of a snakebite for inclusion in the study? This is of utmost importance in case of death.

b. It is stated that the field worker will collect the data from “the victim within 2 weeks of discharge from hospital or from close relatives in case of death”. Again, the investigators seem to assume that all victims also access hospital, which is not the case.

c. Why will ASHA be given incentive for reporting incidence or death due to snakebites in that area, when she is being informed of this by field workers?

d. Who are these field workers? Will they be recruited specifically for this study? What training will they be provided with for completing the tool?

e. Will the Annexures be translated into local languages? Please provide the details of the process.

6. Methodology, data entry (pages 16) - It is stated that the “data gathered at the Panchayat (village) level would be collated on the Questionnaire sheet in the regional language by the field officers, gathered data is then entered on the Tab PC by the District coordinator in English”.

Are these village level data in addition to Annexure 2? Please clarify. Also, please indicate what will be captured in regional language as Annexure 2 seems to be yes/no type of tool.

7. Study challenges (page 18) – Please provide evidence to justify that there will be recall bias in case of snakebite.

8. Annexure 1, the Participant information sheet and Informed Consent

a. The form does not state which organisation is undertaking the research study/data collection. Only names of PI are written with no institutional affiliation.

b. This statement in background is incorrect - “There is no real epidemiological data on the prevalence, incidence and morbidity resulting from snakebite in India”. Please update to reflect the current scenario or simply mention that more specific data are needed.

c. The form will be used for family of the deceased and also for victim who survived. It does not seem to indicate empathy/sympathy for the loss of a family member.

d. The form does not mention the study area, and how the ASHA was able to get the information on snakebite to contact the household for interview.

e. The form does not specify what type of data will be collected from the participants for them to make an informed choice to participate.

f. Informed consent shows space for signature of the Principal Investigator. It is unlikely for him/her to be available at the time of interview.

g. Such forms typically have footer or header to indicate the number of pages for the participant to be aware that nothing is missed/not shown before he/she decides to sign. This is not seen in the annexure provided.

9. Annexure showing the budget is not needed. Please remove.

Journal Requirements:

4. We note that Figures in 'Annexure 3 Topographical maps' in your submission contain map/satellite images which may be copyrighted. All PLOS content is published under the Creative Commons Attribution License (CC BY 4.0), which means that the manuscript, images, and Supporting Information files will be freely available online, and any third party is permitted to access, download, copy, distribute, and use these materials in any way, even commercially, with proper attribution. For these reasons, we cannot publish previously copyrighted maps or satellite images created using proprietary data, such as Google software (Google Maps, Street View, and Earth). For more information, see our copyright guidelines: http://journals.plos.org/plosone/s/licenses-and-copyright.

4.1.    You may seek permission from the original copyright holder of 'Annexure 3 Topographical maps' to publish the content specifically under the CC BY 4.0 license. 

4.2.    If you are unable to obtain permission from the original copyright holder to publish these figures under the CC BY 4.0 license or if the copyright holder’s requirements are incompatible with the CC BY 4.0 license, please either i) remove the figure or ii) supply a replacement figure that complies with the CC BY 4.0 license. Please check copyright information on all replacement figures and update the figure caption with source information. If applicable, please specify in the figure caption text when a figure is similar but not identical to the original image and is therefore for illustrative purposes only.

5. We note you have included a table to which you do not refer in the text of your manuscript. Please ensure that you refer to Table 2 in your text; if accepted, production will need this reference to link the reader to the Table.

Reviewers' comments:

Reviewer's Responses to Questions

**Comments to the Author**

1. Does the manuscript provide a valid rationale for the proposed study, with clearly identified and justified research questions?

Reviewer #1: No

Reviewer #2: Yes

Reviewer #3: Yes

2. Is the protocol technically sound and planned in a manner that will lead to a meaningful outcome and allow testing the stated hypotheses?

Reviewer #1: No

Reviewer #2: No

Reviewer #3: Partly

3. Is the methodology feasible and described in sufficient detail to allow the work to be replicable?

Reviewer #1: No

Reviewer #2: No

Reviewer #3: No

4. Have the authors described where all data underlying the findings will be made available when the study is complete?

Reviewer #1: No

Reviewer #2: No

Reviewer #3: No

5. Is the manuscript presented in an intelligible fashion and written in standard English?

Reviewer #1: No

Reviewer #2: No

Reviewer #3: Yes

6. Review Comments to the Author

You may also provide optional suggestions and comments to authors that they might find helpful in planning their study.

Reviewer #1: Unfortunately, I have to recommend to reject the manuscript. The current document is a study protocol to be used to inform study personnel and to plan the study. It provides general information, which are unspecific and in part redundant (some sentences are even included twice ...). It is not clear who is the target audience of this document, since it does not provide new information relevant to anyone outside the study team.

However, I would like to motivate you to complete the study and to publish the findings as they will be very relevant for the management of snakebites in India. The submitted document can be used to draft parts of the method section.

Reviewer #2: ICMR Task Force Project- Survey of the incidence, mortality, morbidity and socioeconomic

burden of snakebite in India: A study protocol

Major comments

• Sri Lankan study is a community-based study where the study was planned to physically visit and obtained snakebite data from 1% of households in Sri Lanka

• Authors claim this study will cover 6.12% and compared with the above Sri Lankan study. However, this cannot be compared as the proposed study is not a community-based survey with physically visiting houses.

• This study will not be able to accurately estimate the estimate the incidence, mortality, morbidity and socioeconomic burden of snakebite as the snakebite reporting mechanism can easily under report snakebites as the study entirely depend on the reporting of field officer (i.e. not all snakebites will be captures as this is not a community based survey).

• Not clear how the management data (eg. Number of ASV vials) can be obtained without reviewing hospital records as the questionnaire will be filled at households.

• Further, Districts, Taluks, ect are not randomly selected, hence results cannot be generalized to the country.

Minor comments

• Typos and formatting errors in the proposal.

• Some abbreviations are expanded – Eg. ASHA, AIMS

Reviewer #3: This proposal presents a protocol followed in a comprehensive community-based epidemiological assessment of snakebite in 31 districts in India. To achieve the challenge of detailing epidemiological aspects in one of the most densely populated areas affected by snakebite, the authors require a protocol that allows collecting information on a large scale and a good data curation (and conformation) system. Thus, presenting a well-detailed protocol for a community-based study of this nature constitutes an exciting contribution that can help reproduce similar efforts in other settings. As such, it seems to me that the authors should be motivated to disclose it. However, before this is published in PLOs One, I think the authors should address some of the observations below.

Introduction. This section is somewhat long and could be shortened a bit, especially if we consider that the situation of snakebite in India has been previously reported. The authors should recognize that other efforts at the regional and local levels have been carried out in the subcontinent. Thanks to these studies, we have a basis for the complicated situation in India. The justification for this new examination should concentrate on the thoroughness of the population/region covered.

The introduction needs to address a core aspect of the protocol: the organization of health services in the study region. In particular, the definition and role of ASHAs (Accredited Social Health Activists) in health care. These community activists are a peculiarity of the Indian system and vital to collecting epidemiological information in this protocol. Therefore, the mechanisms they use to detect health issues in their communities should be better explained.

Goals. The objectives are reasonable. Those related to the estimation of incidence and mortality and the type of medical care (traditional or modern) followed by those affected can be directly derived from the interviews. Other objectives, such as documenting clinical information (predominant symptoms, duration of hospital stay, vials of ASV, complications etc) will require verification since the patient or her family members are not always aware of the procedures or pathologies suffered. The proposed methodology does not clarify whether this information will come directly from the medical centers or if it will only depend on the people interviewed in the community. If the latter, what would be the mechanisms for confirming the accuracy of that information?

In the Study design section, mention is made of the four most important species from the medical point of view of India (the big four) and other species that, due to the location of the study, could cause some of the recorded accidents. However, the text is long and does not provide more information on the study's design, so I suggest that it be omitted.

Outcomes. A series of possible primary products of this intervention are indicated: on the one hand, epidemiological information on the incidence of snakebite at the geographical level and the estimation of mortality and economic burden. As secondary outcomes, aspects of the poisoning and its treatment are indicated. As mentioned above, it is impossible to determine the origin of the information on clinical characteristics of the treatment in this protocol, which leaves us without knowing how some of these intended outcomes can be achieved. This section is also repetitive.

The proposed methodology states that the survey will be carried out in a region comprising 31 districts, with close to 84 million inhabitants (about 17 million households). Indeed conducting a study of such magnitude at the community level will require an army of participants, including interviewers and organizers. Field workers will conduct interviews, an unspoken number of them every 10-15 blocks. If 336 blocks are expected to be evaluated, then the minimum number of interviewers would be 22, but the actual number is unknown.

The Data Collection, quality checks, and monitoring section represent the core aspects of the protocol. However, some details should be clarified as well. It is indicated that these workers would attend the calls of the ASHAs, who would report the cases directly from the community. As mentioned before, further clarification is required about this alert system and the role that ASHAs play in public health, and the mechanisms of their participation. One aspect mentioned is that ASHAs will receive incentives to report information on snakebite incidence or death. This could incentivize false reports. What control do you have for it?

The field-worker will collect the data on the bite and the treatment. In our own experience, relying on medical information supplied by patients can lead to inaccurate records that lead to erroneous conclusions. How will the authors ensure the quality of the information provided? Although the questionnaires are reviewed to ensure they are complete, there is no indication on how to endorse the quality of the information provided. This aspect is key to distinguish perception studies from those that try to reflect the reality of a situation.

Finally, several of the acronyms used in this section are not explained, which makes revision difficult.

Analytical aspects. Although some of the variables to be evaluated are mentioned, only general references are made to the analytical procedures ("Data collected in the questionnaire will be analyzed using appropriate statistical techniques."), So this component cannot be properly evaluated.

The economic and social burdens will be evaluated from the information provided by the patients, but again there is no indication of verification of the information provided.

In summary, epidemiological studies based on community-level assessments are an essential alternative to collect valuable information on snakebite in situations where an adequate hospital registration system is not available or where the population does not always use the health system to be cared for. The proposal of the study proposed by Menon et al can constitute a very important contribution in this field. However, the submitted manuscript must be trimmed in some repetitive aspects and better detailed in core aspects of data collection and curation. One suggestion to attach the questionnaire presented to participants.

I want to end with a note instead addressed to the editor. As there are no numbered lines in the text, it isn't easy to refer to observations and support suggestions. The magazine's editorial board should consider the numbering of lines as a requirement when submitting the article.

7. PLOS authors have the option to publish the peer review history of their article (what does this mean?). If published, this will include your full peer review and any attached files.

Reviewer #1: No

Reviewer #2: No

Reviewer #3: No

---

## [Author Response · Author response to Decision Letter 0]

29 Aug 2021

REBUTTAL TO THE COMMENTS OF THE REVIEWERS

Additional Editor’s comments-

1. We have modified the manuscript to the journals requirements and hope the revised draft reads better

2. As mentioned in the Introduction, prior studies suggest that only 20-30% of individuals of snakebite visit a modern medicine practitioner while the rest mostly depend on alternate cures available readily in most villages. Roughly two-thirds of bites are non-venomous and half the rest from venomous species are “dry” (no envenoming), which lends credibility to the alternate cures as only about 15-20% of all snakebites lead to envenoming. The basis of planning this as a community based study was so as to be able to assess numbers from the community itself and thereby be able to capture details of snakebites treated by alternate systems as well.

As mentioned details of envenoming and sequel would be captured in all cases irrespective of whether hospitalised or not as given in the Questionnaire sheet.

3. There has not been a study on the incidence and socio-economic burden of snakebite in India the only data available on a pan-India basis (1-MDS) is mortality data only (1-MDS study covered 13 states and not the whole country). The pattern of injuries would be as per the major complications and sequel as generally encountered with venomous snakebite which we think has been adequately addressed. The loss of wages is calculated for both the victim and the care-giver. Cost of treatment calculation used is the same for both fatal and non-fatal victims.

4. As mentioned this is a community based study and course in hospital and treatment may not be uniformly available. What would be entered is the number of vials of ASV used and treatment of complications vis a vis dialysis, ventilator support, use of blood products etc. Q-19 includes Alternate treatment and tells us about the treatment behaviour. As per the suggestion of the reviewer a additional Question has been added to the appended Questionnaire sheet (Date and time of the victim presenting to the Alternate treatment source).

5 a. As mentioned in the section Data collection, monitoring and checks. Identification of victims of snakebite at the ward level would be by ASHAs of the ward and the interview is to be conducted by the Field Officers. One of the exclusion criteria is non-ophid bites and snakebite would be ascertained from the interview of victims, symptoms, fang marks etc and records in case admitted,the same applying to cases of death. 

b. Has been corrected to two weeks of bite or discharge from hospital / death.

c. ASHAs identify the victims for field-officers to conduct the interview.

d. The field-officers are employed specifically for the study recruiting as per ICMR norms for field-officers. Training is imparted on the data entry, common questions asked, symptoms, common venomous snakes, preventive and first-aid measures to be followed in case of bite. A half day training session is done for the field officers of the district with a follow up training after 2 weeks. The same has been done for the states of Rajasthan and Kerala. This is alongside sessions for ASHA workers of the district on the common snakes, signs and symptoms etc. The ASHA training is at the Block / Taluk level and hence multiple sessions for ASHAs are required for a single district. These sessions are in the vernacular with the questionnaire sheets, consent sheets etc being in the local vernacular. 

e. The Questionnaire and Consent sheets are in the local vernacular and have been forward and backward translated by linguists. 

6. ASHAs identify victims of snakebite in their respective wards of the study districts, followed by interviews of the victims by the Field Officer in a Printed Questionnaire sheet which is in the vernacular, the entered details are then entered in English on the Tab provided to the district coordinator for the same which gets stored on the server in AIMS, Kochi the nodal centre for the study. The fields for entry on the Tab PC are the same as given in the Questionnaire other than for being in English.

7. The likelihood of recall bias in those envenomed in unlikely but likely in non-venomous bites.

8. Thank you, the same has been corrected and affiliations added

 b. the wording has been changed to read, there is not adequate data on the incidence and socioeconomic burden of snakebite on a national level in India.

c. I agree with the reviewer in that the Questionnaire does not empathise with the family of the victim in case of death, the field officers/ASHAs will be trained to educate the family/community so as to help prevent further incidents with simple easy measures. Necessary changes have been mada.

d. ASHAs being residents of the ward that they represent typical 1 ASHA for a 1000 population are generally aware of the health status both chronic and emergent conditions of residents of the ward that they serve. 

e. Thank you the same has been corrected.

f. The same has been corrected

g. Thank you the same has been incorporated.

9. Budget has been removed

Reviewer 2 #

As a reply to the major comments:

The proposed study is a community based study in which victims of snakebite are identified by frontline health workers (FLHW) in their respective wards. Wards are the smallest administrative sub-units of a village (panchayat), town or corporation. Each ward elects a member to the village / town council, likewise each ward has a frontline health worker (ASHA- Accredited Social Health Activist) as representative for the health sector at the ward level. On an average there is 1 ASHA to a population of 1000. FLHWs work as the eyes and ears of the health system with an ASHA to a ward (smallest administrative subunit of the village or town). The duties of ASHAs include and not limited to immunisation and family planning, facilitating directly observed therapy short-course (DOTS) therapy for tuberculosis at ward level, surveillance and monitoring for communicable diseases- polio, vector borne febrile illnesses etc, surveillance for NCDs at the ward level, sanitation related activity, participation and facilitation of state or district specific activities at the community level. 

In the present study we use FLHW to identify victims who are then interviewed by the field officer. The interview is entered on the Questionnaire sheet in the vernacular (copy in English –Annexure 2). The gathered details are then entered in English on Tab PCs by the district coordinator which gets synced to the server. 

As mentioned this being a community based study and given that FLHWs are well aware of the health incidences in their respective wards we are confident most incidences would be captured.

As hospitalisation is required for administration of ASV discharge summaries / death summaries of the victims are used for the treatment details including complications/ duration of stay/ vials of ASV etc.

We agree to the reviewer’s comments that the taluks and districts were not randomly selected and cannot be generalised to the country as a whole, but the districts have been picked so as to be representative of the state.

Minor : ASHAs, AIMS etc have been expanded, the same has been corrected, thank you

Reviewer 3 #

Thank you for your comments

We have modified the manuscript as per your suggestions, considerably shortening the Introduction. We have detailed in the Methods section about collection of data from at the community level using Frontline health worker for the same. Treatment data would be as gathered from the discharge / death summary. 

The section on the Big 4 and relevant portions have been omitted all together. 

The data collection section highlighting the role of ASHA for the present proposal and their other duties have also been re-written. The district coordinators and PIs of the district are tasked with verification of entered data by way of phone calls. Random verification of data entered and gathered through the ASHA workers and field officers would be done by the District Coordinator and PI of the district.

---

## [Decision Letter · Decision Letter 1]

25 Mar 2022

PONE-D-21-07087R1ICMR Task Force Project- Survey of the incidence, mortality, morbidity and socio-economic burden of snakebite in India: A study protocolPLOS ONE

Dear Dr. MENON,

Thank you for submitting your manuscript to PLOS ONE. After careful consideration, we feel that it has merit but does not fully meet PLOS ONE’s publication criteria as it currently stands. Therefore, we invite you to submit a revised version of the manuscript that addresses the points raised during the review process.

We look forward to receiving your revised manuscript.

Kind regards,

Benito Soto-Blanco, DVM, MSc, PhD

Academic Editor

PLOS ONE

Journal Requirements:

Reviewers' comments:

Reviewer's Responses to Questions

**Comments to the Author**

1. Does the manuscript provide a valid rationale for the proposed study, with clearly identified and justified research questions?

Reviewer #3: Yes

Reviewer #4: Yes

Reviewer #5: Yes

Reviewer #6: Yes

Reviewer #7: Yes

2. Is the protocol technically sound and planned in a manner that will lead to a meaningful outcome and allow testing the stated hypotheses?

Reviewer #3: Yes

Reviewer #4: Partly

Reviewer #5: Yes

Reviewer #6: Partly

Reviewer #7: Partly

3. Is the methodology feasible and described in sufficient detail to allow the work to be replicable?

Reviewer #3: Yes

Reviewer #4: No

Reviewer #5: Yes

Reviewer #6: No

Reviewer #7: Yes

4. Have the authors described where all data underlying the findings will be made available when the study is complete?

Reviewer #3: Yes

Reviewer #4: Yes

Reviewer #5: Yes

Reviewer #6: No

Reviewer #7: Yes

5. Is the manuscript presented in an intelligible fashion and written in standard English?

Reviewer #3: Yes

Reviewer #4: Yes

Reviewer #5: Yes

Reviewer #6: Yes

Reviewer #7: Yes

6. Review Comments to the Author

You may also provide optional suggestions and comments to authors that they might find helpful in planning their study.

Reviewer #3: This new version of the protocol is more precise and concise, showing the procedure for a massive effort to gather information on the snakebite in India from the perception of those affected. I believe that the modifications made by the authors in this new version are adequate, and the protocol can be published.

Reviewer #4: Thank you very much for inviting me to review this important study protocol to study the incidence, mortality, morbidity and socio-economic burden of snakebite in India. Overall I would like to congratulate the authors on developing a comprehensive protocol. However, there are few major concerns, especially with regard to the sampling strategy and data collection, which need to be addressed by the authors for the data collected to be accurate and reflective of the actual burden of snakebites in the population studied.

Study aims

Should be corrected as ‘Our study aims to estimate the epidemiological and socio-economic effects due to snakebite in India’

Study design

The authors need to give more details of the sampling strategy including how the individual blocks are selected within each districts, how many wards are roughly included in a block etc. It would be clearer if the authors could use a flow diagram to show the sampling strategy.

Data collection

There major concerns with regard to the adequacy and accuracy of the data collection.

1. The authors state that the ASHAs will cover each ward that will include 1000 population. Is there a possibility that ASHAs may miss some of the snakebites that occur? If so what are the measures the investigators have taken to minimize missed reporting? Have they thought of carrying out public awareness campaigns about informing about the snakebites to their ASHAs before the study is started?

2. The authors need to provide more information on the gathering of information and filling the questionnaires by the field workers. It appears that most of the information will be obtained through hospital discharge notes. What happens if there are no or incomplete discharge summaries? How are the investigators going to obtain information about individuals who have sought native/ local treatment where there will be no treatment records?

3. One of the main concerns of the study protocol is the accurate identification of the snake species. They need to provide more details on how this will be done. Are they going to rely on hospital records for this? Are they going to limit species identification to cases where the dead or the live snake was brought to the hospital? Will they include species identified based on the description of the snake?

Reviewer #5: Good study proposal on an epidemiological topic that appears to have a dearth of current literature. Challenges that likely will arise include lack of confirmation of snake, recall bias, dry bites, etc. Would recommend keeping working data evaluating whether the snake was visualized, how the confirmation took place (photo, visual, combined with consistent clinical findings) to get a better sense of distribution. This could also help inform the geographic distribution patterns of specific species in different areas. This project will be a big undertaking, however is worthwhile and ambitious. Determining socioeconomic burden may prove challenging also as metrics will need to be normalized across various income levels, etc. May be worthwhile tabulating total costs of treatment as an objective figure to get a sense of the baseline findings. Are the median income levels and socioeconomic determinants of each geographical area/district being considered? Overall, this is an interesting study and I look forward to the findings.

Reviewer #6: 

- important study but I had a few comments/concerns:

- the study design - is it prospective, a survey, or retrospective? It is said to be prospective, but recall bias is mentioned which would be an issue for a retrospective design?

- there is use of “his” in referring to investigators but “her” when referring to the site workers (ASHWs) - this needs to be changed to gender neutral language (eg their)

- acronyms need to be spelt out in full at first time of use - eg ASHW, NCD

- in abstract the word “ward” is used, but this is an ambiguous term - in some countries a “ward” in an inpatient unit, and so I originally interpreted this as inpatient recruitment only, and hence big flaws in capturing true incidence. Later on in paper it is defined as a small community village unit. Suggest abstract word is changed to a non-ambiguous term

- towards the end section there is use of the term “down trodden” poor people - I think socioeconomically disadvantaged probably a better/preferred term.

Thanks for invitation.

Reviewer #7: The authors present interesting data on a largely neglected area of research, i.e. envenoming epidemiology. The scarcity of data in this field necessitates that any data that can be published, should be published and, thus, I believe this study merits publication. However, there remain some major revisions to ensure the coherency and soundness of the presented data.

7. PLOS authors have the option to publish the peer review history of their article (what does this mean?). If published, this will include your full peer review and any attached files.

Reviewer #3: **Yes: **Mahmood Sasa Marin

Reviewer #4: **Yes: **Eranga Sanjeewa Wijewickrama

Reviewer #5: No

Reviewer #6: No

Reviewer #7: No

---

## [Author Response · Author response to Decision Letter 1]

19 May 2022

Reviewer #3: This new version of the protocol is more precise and concise, showing the procedure for a massive effort to gather information on the snakebite in India from the perception of those affected. I believe that the modifications made by the authors in this new version are adequate, and the protocol can be published.

Thank you for your comment. 

Reviewer #4: Thank you very much for inviting me to review this important study protocol to study the incidence, mortality, morbidity and socio-economic burden of snakebite in India. Overall I would like to congratulate the authors on developing a comprehensive protocol. However, there are few major concerns, especially with regard to the sampling strategy and data collection, which need to be addressed by the authors for the data collected to be accurate and reflective of the actual burden of snakebites in the population studied.

Study aims

Should be corrected as ‘Our study aims to estimate the epidemiological and socio-economic effects due to snakebite in India’

Response: Thank you, the same has been corrected.

Study design

The authors need to give more details of the sampling strategy including how the individual blocks are selected within each districts, how many wards are roughly included in a block etc. It would be clearer if the authors could use a flow diagram to show the sampling strategy.

Response: The following text has now been inserted in Study Design section

The cross-sectional population in this case is the study population comprising all resident individuals in the study district. 

Data collection

There major concerns with regard to the adequacy and accuracy of the data collection.

1. The authors state that the ASHAs will cover each ward that will include 1000 population. Is there a possibility that ASHAs may miss some of the snakebites that occur? If so what are the measures the investigators have taken to minimize missed reporting? Have they thought of carrying out public awareness campaigns about informing about the snakebites to their ASHAs before the study is started?

Response: Following text has now been inserted in the manuscript.

We are aware that there would be instances where ASHAs would might miss some of the snakebites that occur. These will the minimized to a large extent through the following; training sessions for ASHAs for identification of any warning symptoms in case of a bite, do’s and dont’s following a bite, and identification of the common venomous snakes of the region. These training sessions will also include use of Google Forms using tablets for ASHAs to enter the data as mentioned in Annexure 2. Additionally, through the grant, we have also kept a provision for incentivising the ASHAs for reporting each snakebite case to the District Coordinator. 

2. The authors need to provide more information on the gathering of information and filling the questionnaires by the field workers. It appears that most of the information will be obtained through hospital discharge notes. What happens if there are no or incomplete discharge summaries? How are the investigators going to obtain information about individuals who have sought native/ local treatment where there will be no treatment records?

Response: Following text has now been inserted into the manuscript.

In cases where the discharge/death summaries are incomplete, the District Coordinator will be informed who will coordinate with the Principal Investigator of the state to reach out to the hospital/doctor to receive all relevant details. For the cases where the victim has sought native or local treatment, this will be documented by the ASHAs based on discussions with the victim, and will be supplemented by any additional information by the District Coordinator wherever needed.

3. One of the main concerns of the study protocol is the accurate identification of the snake species. They need to provide more details on how this will be done. Are they going to rely on hospital records for this? Are they going to limit species identification to cases where the dead or the live snake was brought to the hospital? Will they include species identified based on the description of the snake?

Response: This study is not designed to identify the biting species. The drop down menu in the questionnaire sheet would show a broad division into hematoxic and neurotoxic based on symptoms / discharge records (in those admitted). We have given a provision for entering the photograph of the snake if available. The study is basically designed for the incidence, mortality, morbidity and not as much for the identification of the snake species. 

Reviewer #5: Good study proposal on an epidemiological topic that appears to have a dearth of current literature. Challenges that likely will arise include lack of confirmation of snake, recall bias, dry bites, etc. Would recommend keeping working data evaluating whether the snake was visualized, how the confirmation took place (photo, visual, combined with consistent clinical findings) to get a better sense of distribution. This could also help inform the geographic distribution patterns of specific species in different areas. This project will be a big undertaking, however is worthwhile and ambitious. Determining socioeconomic burden may prove challenging also as metrics will need to be normalized across various income levels, etc. May be worthwhile tabulating total costs of treatment as an objective figure to get a sense of the baseline findings. Are the median income levels and socioeconomic determinants of each geographical area/district being considered? Overall, this is an interesting study and I look forward to the findings.

Response: Thank you for your comments. This study is not designed to identify the biting species. The drop down in the questionnaire sheet would show a broad division into hematoxic and neurotoxic based on symptoms / discharge records (in those admitted). We have also given a provision for entering the photograph of the snake if available. The study is basically designed for the incidence, mortality, morbidity and not as much for snake species identification. 

Yes, the income of the household is being considered. There is a provision for upload of the bitten part / snake if available. All valuable suggestions which have been considered. The other social determinants like educational background will also be considered. 

Reviewer #6: 

- important study but I had a few comments/concerns:

- the study design - is it prospective, a survey, or retrospective? It is said to be prospective, but recall bias is mentioned which would be an issue for a retrospective design?

Response: This is a prospective cross-sectional study study on which we look at an event of snakebite (outcome) in the cohort of a defined population (i.e. the entire population of the district) reporting incidence of snakebite. The data is to be reported within 2 weeks and the mention of recall bias was in relation to victims forgetting about the incident especially in cases of dry/ non-venomous bites, which is why the window was kept at 2 weeks. 

- there is use of “his” in referring to investigators but “her” when referring to the site workers (ASHWs) - this needs to be changed to gender neutral language (eg their)

- acronyms need to be spelt out in full at first time of use - eg ASHW, NCD

- in abstract the word “ward” is used, but this is an ambiguous term - in some countries a “ward” in an inpatient unit, and so I originally interpreted this as inpatient recruitment only, and hence big flaws in capturing true incidence. Later on in paper it is defined as a small community village unit. Suggest abstract word is changed to a non-ambiguous term

- towards the end section there is use of the term “down trodden” poor people - I think socioeconomically disadvantaged probably a better/preferred term.

Response: Thank you, the same has been corrected and explained as suggested by the reviewer. 

Thanks for invitation.

Reviewer #7: The authors present interesting data on a largely neglected area of research, i.e. envenoming epidemiology. The scarcity of data in this field necessitates that any data that can be published, should be published and, thus, I believe this study merits publication. However, there remain some major revisions to ensure the coherency and soundness of the presented data.

Response: We have now addressed al the comments by reviewers and incorporated them in the revised version. We would like thank the reviewers for their comments as we believed that it has helped make the protocol better. 

General 

Thank you for noting that your data will be made available. Please note that PLOS ONE’s policies on Data Availability require that all data necessary to replicate the findings of the manuscript be made available upon acceptance.

PLOS ONE strongly recommends that this data be deposited to a stable public repository such as Figshare (http://figshare.com/) or Dryad (datadryad.org). For a list of other PLOS recommended repositories, please visit: http://journals.plos.org/plosone/s/data-availability#loc-recommended-repositories.

Response: Annexures used in this protocol are available in public repository atJohn, Denny; Menon, Jaideep (2021): ICMR –Task Force Study on the incidence, mortality, morbidity and socio-economic burden of Snakebite in India: A study protocol. figshare. Dataset. https://doi.org/10.6084/m9.figshare.14072108.v1

---

## [Decision Letter · Decision Letter 2]

17 Jun 2022

ICMR Task Force Project- Survey of the incidence, mortality, morbidity and socio-economic burden of snakebite in India: A study protocol

PONE-D-21-07087R2

Dear Dr. MENON,

We’re pleased to inform you that your manuscript has been judged scientifically suitable for publication and will be formally accepted for publication once it meets all outstanding technical requirements.

Kind regards,

Benito Soto-Blanco, DVM, MSc, PhD

Academic Editor

PLOS ONE

Reviewers' comments:

Reviewer's Responses to Questions

**Comments to the Author**

1. Does the manuscript provide a valid rationale for the proposed study, with clearly identified and justified research questions?

Reviewer #4: Yes

2. Is the protocol technically sound and planned in a manner that will lead to a meaningful outcome and allow testing the stated hypotheses?

Reviewer #4: Yes

3. Is the methodology feasible and described in sufficient detail to allow the work to be replicable?

Reviewer #4: Yes

4. Have the authors described where all data underlying the findings will be made available when the study is complete?

Reviewer #4: Yes

5. Is the manuscript presented in an intelligible fashion and written in standard English?

Reviewer #4: Yes

6. Review Comments to the Author

You may also provide optional suggestions and comments to authors that they might find helpful in planning their study.

Reviewer #4: Thank you very much for letting me review the revised version of this study protocol. The authors have adequately addressed the concerns raised during the review. I recommend publication of this study protocol in PLOS One.

7. PLOS authors have the option to publish the peer review history of their article (what does this mean?). If published, this will include your full peer review and any attached files.

Reviewer #4: **Yes: **Eranga Sanjeewa Wijewickrama

---

## [Editor Report · Acceptance letter]

10 Jul 2022

PONE-D-21-07087R2 

ICMR Task Force Project- Survey of the incidence, mortality, morbidity and socio-economic burden of snakebite in India: A study protocol 

Dear Dr. Menon:

I'm pleased to inform you that your manuscript has been deemed suitable for publication in PLOS ONE. Congratulations! Your manuscript is now with our production department. 

Kind regards, 

on behalf of

Dr. Benito Soto-Blanco 

Academic Editor

PLOS ONE